# Real-World Outcomes of Chemoradiotherapy in Patients with Stage II/III Non-Small-Cell Lung Cancer in the Durvalumab Era: An Observational Study

**DOI:** 10.3390/cancers17152498

**Published:** 2025-07-29

**Authors:** Jörg Andreas Müller, Jonas Buchberger, Elias Schmidt-Riese, Clara Pitzschel, Miriam Möller, Wolfgang Schütte, Daniel Medenwald, Dirk Vordermark

**Affiliations:** 1Department of Radiation Oncology, University Hospital Halle (Saale), Ernst-Grube-Str. 40, 06120 Halle (Saale), Germany; jonasbuchberger@gmx.de (J.B.); elias.schmidt-riese@uk-halle.de (E.S.-R.); clara.pitzschel@uk-halle.de (C.P.); publikationen.stra-med@med.ovgu.de (D.M.); dirk.vordermark@uk-halle.de (D.V.); 2Institute of Computer Science, Martin-Luther-University Halle-Wittenberg, Von-Seckendorff-Platz 1, 06120 Halle (Saale), Germany; 3Second Medical Department, City Hospital Martha-Maria Halle-Doelau, 06120 Halle (Saale), Germany; miriam.moeller@martha-maria.de (M.M.); wolfgang.schuette@martha-maria.de (W.S.); 4University Clinic for Radiation Therapy, University Hospital Magdeburg A. ö. R, 39120 Magdeburg, Germany

**Keywords:** stage III NSCLC, durvalumab era, real-world outcome, immunotherapy, lung cancer

## Abstract

Durvalumab is a drug used after chemoradiotherapy (CRT) to help the immune system fight lung cancer. While clinical trials like the PACIFIC study showed that this approach improves survival in patients with stage II/III non-small-cell lung cancer (NSCLC), it is unclear how well these results apply to everyday clinical practice. In our study, we looked at real-world outcomes in 72 patients treated at a single center in Germany. We found that patients who received durvalumab after CRT lived longer and had fewer disease progressions—especially those with good general health and few other illnesses. These findings confirm that durvalumab works well outside of clinical trials and highlight the importance of overall fitness and low comorbidity in lung cancer treatment.

## 1. Introduction

Non-small-cell lung cancer (NSCLC) represents approximately 85% of all lung cancer cases and remains a leading cause of cancer-related mortality. While recent registry data indicate a 5-year survival rate of 26.4% across all stages [1], real-world evidence on outcomes in stage III NSCLC—particularly regarding multimodal treatment and the integration of consolidation immunotherapy—remains limited.

Immune checkpoint inhibitors targeting programmed death 1 (PD-1) and programmed death ligand 1 (PD-L1) have demonstrated clinical efficacy across various tumor entities, including advanced NSCLC [2,3]. Durvalumab, a selective, high-affinity, human IgG1 monoclonal antibody, blocks the interaction of PD-L1 with both PD-1 and CD80, thereby enhancing T-cell-mediated antitumor activity [4,5]. Preclinical studies indicate that concurrent chemoradiotherapy can induce upregulation of PD-L1 expression in tumor cells. This has led to the hypothesis that subsequent PD-L1 inhibition may restore and sustain systemic immune responses, potentially improving long-term tumor control after chemoradiotherapy (CRT) [5,6,7,8,9,10,11].

The PACIFIC trial established durvalumab consolidation as a new standard of care for patients with unresectable stage III NSCLC who show no progression after platinum-based CRT. Durvalumab significantly improved progression-free survival (PFS) (median 16.8 vs. 5.6 months; HR 0.52) and extended the median time to death or distant metastasis (23.2 vs. 14.6 months), indicating a delayed onset of systemic disease progression. These benefits were achieved with a manageable safety profile and led to a paradigm shift in the treatment of stage III NSCLC [11]. Building on the initial PACIFIC trial results, the 3-year follow-up data confirmed the long-term survival benefit of durvalumab following CRT in patients with unresectable stage III NSCLC. The 3-year overall survival rate was markedly higher in the durvalumab group compared to placebo (57.0% vs. 43.5%), reinforcing the role of durvalumab as a standard of care in this setting. These results underscore the durable efficacy of consolidation immunotherapy and highlight its potential to significantly alter the prognosis of this patient population [12]. The findings from the PACIFIC-R study by Filippi, Bar et al. further reinforce the real-world effectiveness of consolidation durvalumab following chemoradiotherapy in unresectable stage III NSCLC. With a 3-year overall survival rate of 63.2%, these results closely mirror the survival benefit observed in the PACIFIC trial. Notably, the study demonstrated improved outcomes in patients with PD-L1 expression ≥ 1% and those treated with concurrent CRT, yet also reported favorable survival rates among patients with lower PD-L1 expression and sequential CRT. These data highlight the broad applicability of durvalumab across a heterogeneous real-world patient population and confirm its role as a standard post-CRT treatment [13].

Building upon the evidence from PACIFIC and PACIFIC-R, real-world data offer valuable insights into the effectiveness and safety of consolidation durvalumab in more diverse and less-selected patient populations. A recent meta-analysis of 13 real-world studies including 1,885 patients demonstrated that, despite broader inclusion of older patients, those with poorer performance status, and variations in treatment timing, short-term outcomes remained consistent with the PACIFIC trial. Specifically, the pooled 12-month OS and PFS rates were 90% and 62%, respectively, while the safety profile remained acceptable, with grade ≥ 3 pneumonitis occurring in only 6% of patients. These findings confirm that the clinical benefits of durvalumab observed in controlled trials are largely reproducible under real-world conditions, further underscoring its role as a standard of care in unresectable stage III NSCLC [14].

Arunachalam, Sura et al. (2024) [15] analyzed 426 patients from U.S. community oncology settings and found a median real-world OS of 50.2 months in the concurrent CRT + durvalumab group versus 11.6 months in the concurrent CRT-alone group. Median real world PFS was 28.5 vs. 6.3 months, respectively, further confirming the substantial survival benefit of durvalumab in real-world practice [15]. The KINDLE-Korea study reported data from a real-world cohort (n = 461) treated prior to durvalumab approval. This analysis highlighted heterogeneity in treatment strategies and outcomes, and a median OS of 66.7 months in patients primarily treated with surgery or CRT. These findings underscore the ongoing need to improve outcomes, particularly for patients not eligible for surgical management [16]. Denault, Feng et al. (2023) conducted a real-world survival analysis stratified by PD-L1 status and found that durvalumab improved OS in the PD-L1 ≥ 1% subgroup (HR 0.53, *p* = 0.003), but not in PD-L1 < 1% patients, suggesting a potential biomarker-based effect modifier, as observed in post hoc PACIFIC analyses [17].

The aim of this study is to provide an additional contribution to the growing body of real-world evidence on outcomes following CRT in the era of consolidation durvalumab. By analyzing patient characteristics, treatment patterns, and survival outcomes in a real-world clinical setting, this work seeks to evaluate the applicability of PACIFIC trial findings to routine practice and to identify factors that may influence prognosis in daily care. This analysis adds valuable insights into the effectiveness of current treatment strategies for patients with unresectable stage III NSCLC outside the context of clinical trials.

## 2. Methods

### 2.1. Data and Material

Patient recruitment was conducted retrospectively using the digital archives of the Department of Radiation Oncology at University Hospital Halle (Saale). Data were anonymized and extracted from the hospital information system ORBIS (version 03.20.02.01, Bonn, Germany). Imaging diagnostics and radiotherapy data were obtained from Centricity PACS (GE Healthcare, version 2.2.8497.22382, Chicago, IL, USA) and Elekta Mosaiq (version 2.84, Stockholm, Sweden). All patients with stage II–III non-small-cell lung cancer (NSCLC) who received definitive concurrent chemoradiotherapy (CRT) between 2017 and 2022 were considered. In line with this real-world approach, patients with higher T-stages (e.g., T3–T4) and nodal involvement (N1–N2) were not excluded from this analysis. For statistical analysis, T-stage and N-stage were simplified into relevant categories (T1–2 vs. T3–4 and N0 vs. N1–N3) to ensure interpretability.

This study received a positive vote and was approved by the ethics committee of the Medical Faculty of Martin Luther University Halle-Wittenberg.

Clinical variables were categorized for statistical analysis. Sex was classified as male or female. Age at diagnosis was treated as a continuous variable. Performance status was assessed using the Karnofsky Performance Status (KPS) and dichotomized into two groups: >80% (indicating good performance) and ≤80% (indicating impaired performance). The Karnofsky Performance Status (KPS) enables clinicians to quantify a patient’s functional capacity, thereby supporting comparative assessments of treatment efficacy and prognostic outcomes. Lower KPS scores are consistently associated with reduced survival across a range of serious medical conditions [18].

Histological subtype was grouped as adenocarcinoma or squamous cell carcinoma.

PD-L1 expression was reported as a percentage and categorized into three groups: <1%, ≥1%, and unknown. The administration of durvalumab was recorded as a binary variable (yes/no). The Charlson Comorbidity Index (CCI) was used to evaluate the burden of comorbidities and was categorized into two groups: ≤2, and >2. The Charlson Comorbidity Index (CCI) is a widely applied instrument for estimating patient prognosis and 10-year survival, based on the presence and severity of comorbid conditions. Table 1 provides an overview of the comorbidities included in the CCI and their respective weighting within the score [19].

Smoking exposure was quantified using pack-years as a continuous variable. Radiation therapy was considered complete if the full prescribed course was delivered; this variable was treated as binary (yes/no).

### 2.2. Statistical Analyses

We used proportional hazard Cox regression models to assess the association of cancer-related parameters with mortality and computed hazard ratios (HRs) with 95% confidence intervals (CI). Progression-free survival (PFS) and overall survival (OS) were calculated from the end of radiotherapy to either disease progression, death, or last follow-up. In cases where the PFS time was missing, it was imputed based on the difference between the end of radiotherapy and the censoring date (28 July 2024), if available.

In addition to a full multivariable model including all clinically relevant covariates, a stepwise model selection approach was employed to derive an optimized, parsimonious model. Stepwise selection was performed using the step() function in R with a bidirectional approach (“both”), iteratively adding or removing variables based on their contribution to model performance.

The primary criterion for model selection was the Akaike Information Criterion (AIC), with lower AIC values indicating a better trade-off between model fit and complexity. In addition, the log-likelihood, Wald test, and likelihood ratio test (LRT) were evaluated to assess overall model quality and the statistical significance of individual predictors. The final model was defined as the one with the lowest AIC and a clinically meaningful combination of variables. A *p*-value of <0.05 was considered statistically significant.

To account for potential confounding due to changing clinical practice and durvalumab availability, all models were adjusted for treatment period (2017–2018 vs. 2019–2022). In addition, a sensitivity analysis using nonparametric bootstrap resampling (1000 iterations) was performed to assess the robustness of the multivariable results.

The statistical analyses were conducted using RStudio, version 2024.04.2+764.

During the preparation of this work, the author(s) used ChatGPT (version GPT-4.1), a language model developed by OpenAI Inc., San Francisco, CA, USA, in order to improve writing style and check grammar and spelling. After using this tool, the authors reviewed and edited the content as needed and take full responsibility for the content of the publication.

## 3. Results

### 3.1. Case Selection

A total of 80 patients with stage II–III non-small-cell lung cancer (NSCLC) who underwent definitive CRT between 2017 and 2022 at Halle (Saale) were identified. After excluding 8 patients with missing information regarding the administration of durvalumab, 72 patients were finally included in the analysis. The patient selection process is summarized in Figure 1.

### 3.2. Patient Characteristics

A total of 72 patients with stage IIb–IIIc non-small-cell lung cancer (NSCLC) who received definitive chemoradiotherapy (CRT) were included in this retrospective analysis. Of these, 35 patients received consolidation therapy with durvalumab following CRT, and 37 patients underwent CRT without durvalumab.

The median age at diagnosis was 66 years (range 46–83), and most patients were male (75%). Median smoking exposure was 34 pack-years (range 0–60). The median CCI was 2.0, and the median Karnofsky performance status was 80% in both groups. Baseline characteristics were generally balanced between the two cohorts.

However, notable differences were observed in PD-L1 expression: 60% of patients in the durvalumab group had PD-L1 levels > 10%, compared to only 14% in the CRT-only group. In contrast, low PD-L1 expression (<1%) was more common in the CRT-only group (51% vs. 2.9%). Furthermore, among patients with PD-L1 expression ≥ 1%, the vast majority received durvalumab (97%), while those with PD-L1 < 1% were primarily treated without durvalumab (51%). Only a small subset of patients with PD-L1 < 1% received durvalumab (2.9%). Patients with unknown PD-L1 status (11%) did not receive durvalumab.

When stratified by treatment period, reflecting the clinical introduction of durvalumab in routine practice starting around 2019, a clear temporal pattern emerged: durvalumab was administered almost exclusively after 2019. From 2017–2019, most patients with PD-L1 ≥ 1% still did not receive durvalumab (12 with, 6 without), whereas from 2020–2022, durvalumab was increasingly used in PD-L1-positive patients (21 with, only 4 without). Similarly, patients with low or unknown PD-L1 expression were rarely treated with durvalumab in either period.

T- and N-stage distributions were comparable across groups, with the majority of patients presenting with T3–T4 tumors and N2–N3 nodal status. Stage III disease accounted for over 95% of cases in both groups, with slightly more patients with stage IIIb in the durvalumab cohort (43% vs. 32%).

Squamous cell carcinoma was the most frequent histologic subtype (74% in the durvalumab group vs. 63% in CRT-only), while adenocarcinoma was more frequent in the CRT-only group (34% vs. 23%). Most patients in both groups received cisplatin/vinorelbine chemotherapy (63% vs. 57%), and the median total radiation dose was 66 Gy in both groups. Table 2 summarizes the essential baseline characteristics of patients treated with CRT, stratified by receipt of consolidation durvalumab. Complete baseline characteristics can be found in Appendix A.

### 3.3. Survival Analyses

The median OS for the entire cohort was 2.08 years. The 3-year and 5-year OS rates were 38.6% and 30.3%, respectively. Patients who received durvalumab had a median OS of 5.00 years, while those who did not receive durvalumab had a median OS of 1.75 years.

Subgroup analysis revealed longer median OS in patients with higher KPS (>80) at 15 months (IQR: 7–>29) compared to those with ≤80 (13 months, IQR: 7–33). Similarly, patients with lower comorbidity (CCI ≤ 2) had a median OS of 15 months (IQR: 9–29) versus 18 months (IQR: 16–41) in patients with CCI > 2, although the latter showed a wider interquartile range suggesting higher variability.

Although PD-L1 expression was not statistically significant in univariable analysis, patients with ≥1% PD-L1 expression had longer median survival (16–18 months) compared to those with <1% (17 months).

In the univariable Cox regression analysis for overall survival, higher Karnofsky performance status (>80) was significantly associated with improved survival (HR 0.42, 95% CI 0.22–0.82, *p* = 0.007). Female sex also showed a trend toward better survival (HR 0.48, 95% CI 0.21–1.06, *p* = 0.049), while other variables such as age at diagnosis, CCI group, T-stage, N-stage, PD-L1 status, and durvalumab treatment did not reach statistical significance.

In the multivariable model derived through stepwise selection, several factors were independently associated with overall survival. A Karnofsky index > 80% remained a strong predictor of improved survival (HR 0.29, 95% CI 0.12–0.69, *p* = 0.003). Similarly, patients with a CCI ≤ 2 had significantly better outcomes compared to those with CCI > 2 (HR 0.39, 95% CI 0.18–0.82, *p* = 0.009). Most notably, receipt of durvalumab was independently associated with a significantly reduced risk of death (HR 3.99, 95% CI 1.48–10.8, *p* = 0.008). PD-L1 expression ≥ 1% also showed a trend toward improved survival compared to <1%, though it did not reach statistical significance (HR 3.72, 95% CI 1.27–10.9, *p* = 0.063). Nodal stage showed borderline significance (*p* = 0.093), while T-stage and age remained non-significant in the final model. A detailed overview of univariable and multivariable Cox regression results is provided in Table 3.

Kaplan–Meier curves were generated to illustrate OS stratified by key clinical factors. Patients who received durvalumab following CRT showed significantly improved survival compared to those who did not (Figure 2). Similarly, patients with a Karnofsky performance status > 80% had superior survival outcomes compared to those with a status ≤ 80% (Figure 3). When stratified by both durvalumab treatment and Karnofsky index, patients with a high performance status and durvalumab therapy showed the most favorable outcomes (Figure 4). Differences between groups were evaluated using the log-rank test.

The median progression-free survival (PFS) for the total cohort was 1.17 years. The estimated 3-year PFS rate was 31.1%, and the 5-year PFS rate was 26.3%. The median PFS was 20.5 months in patients treated with durvalumab compared to 12.0 months in those without durvalumab.

In the univariate model, Karnofsky performance status > 80% was associated with a significantly better PFS (HR 0.37, 95% CI 0.19–0.76, *p* = 0.003), and CCI ≤ 2 showed a nonsignificant trend toward improved outcomes (HR 0.73, 95% CI 0.40–1.34, *p* = 0.3). Other variables, such as age at diagnosis, sex, T-stage, N-stage, PD-L1 status, and durvalumab treatment, were not significantly associated with PFS in univariate models.

In the multivariable Cox regression model selected using a stepwise AIC approach, Karnofsky performance status > 80% (HR 0.29, 95% CI 0.14–0.60, *p* < 0.001), CCI ≤ 2 (HR 0.53, 95% CI 0.28–1.01, *p* = 0.048), and durvalumab treatment (HR 2.81, 95% CI 1.22–6.47, *p* = 0.023) were significantly associated with PFS. PD-L1 expression was also considered in the model (≥1%: HR 1.70, 95% CI 0.72–4.04; unknown: HR 0.40, 95% CI 0.14–1.14; overall *p* = 0.039), while T-stage was not retained. Table 4 presents the results of both univariate and multivariate regression models analyzing PFS.

A Kaplan–Meier analysis indicated a trend toward longer PFS among patients who received durvalumab compared to those who did not, although the difference did not reach statistical significance based on the log-rank test (Figure 5). Additionally, Figure 6 displays the PFS stratified by both durvalumab treatment and Karnofsky performance status, demonstrating that patients with a Karnofsky index > 80 and durvalumab therapy had the most favorable outcomes, highlighting the combined prognostic relevance of performance status and treatment.

### 3.4. Adverse Events

A total of 36 patients (47%) reported treatment-related dysphagia, making it the most common adverse event in the overall cohort (N = 80). Other frequently reported symptoms included cough (30%), dyspnea (24%), erythema (22%), and nausea or vomiting (17%). Data on adverse events were missing in four patients across all categories.

When stratified by treatment group, the incidence of adverse events was generally similar between patients who received durvalumab and those who did not. Dysphagia occurred in 51% of patients in the durvalumab group and 47% in the CRT-only group. Cough was reported in 31% vs. 35%, dyspnea in 20% vs. 29%, erythema in 26% vs. 24%, and nausea/vomiting in 11% vs. 21%, respectively. Although differences in prevalence were observed, these were not statistically analyzed due to the limited sample size. Importantly, adverse events were documented retrospectively without standardized grading (e.g., CTCAE), and only presence or absence was recorded, precluding analysis of severity. A summary of selected adverse events observed during and after CRT, stratified by durvalumab treatment, is provided in Appendix A.

### 3.5. Sensitivity Analysis

To assess the robustness of the primary regression results in light of the introduction of durvalumab in clinical routine after 2019, a sensitivity analysis was conducted. The multivariable Cox regression models for PFS and OS were adjusted for treatment period (2017–2018 vs. 2019–2022) and validated using nonparametric bootstrapping with 1000 iterations.

For progression-free survival (PFS), adjustment for treatment period confirmed the significance of Karnofsky performance status >80 percent (HR 0.29, 95% CI: 0.14 to 0.62, *p* < 0.001), CCI ≤ 2 (HR 0.50, 95% CI: 0.26 to 0.96, *p* = 0.032), and durvalumab treatment (HR 3.75, 95% CI: 1.45 to 9.73, *p* = 0.010). PD-L1 expression ≥ 1 percent was associated with a trend toward worse PFS (HR 2.31, 95% CI: 0.83 to 6.38, *p* = 0.043), while the treatment period itself was not a significant predictor (HR 1.56, 95% CI: 0.68 to 3.59, *p* = 0.3). Bootstrap confidence intervals supported these findings, with Karnofsky > 80 percent (HR 0.29, 95% CI: 0.06 to 0.73) and CCI ≤ 2 (HR 0.39, 95% CI: 0.11 to 0.78) remaining robust.

For overall survival (OS), Karnofsky > 80 percent (HR 0.31, 95% CI: 0.13 to 0.70, *p* = 0.003), CCI ≤ 2 (HR 0.40, 95% CI: 0.20 to 0.82, *p* = 0.009), and durvalumab treatment (HR 4.60, 95% CI: 1.74 to 12.2, *p* = 0.004) remained independently associated with better outcomes. PD-L1 ≥ 1 percent was associated with inferior OS (HR 5.08, 95% CI: 1.64 to 15.8, *p* = 0.024), while the treatment period again did not reach significance (HR 2.25, 95% CI: 0.89 to 5.68, *p* = 0.076). Bootstrap validation yielded consistent hazard ratios, including for durvalumab no vs. yes (HR 4.57, 95% CI: 1.33 to 25.56) and PD-L1 ≥ 1 percent (HR 4.91, 95% CI: 1.44 to 43.37).

Kaplan–Meier estimates adjusted for treatment period are shown in Appendix A, illustrating comparable survival probabilities across both time frames. The corresponding regression results are presented in Appendix A. Collectively, these analyses confirm that the observed associations for PFS and OS, particularly with regard to durvalumab and performance status, are not confounded by changes in treatment availability over time and thus support the robustness of the main findings.

## 4. Discussion

The primary aim of this retrospective observational study was to evaluate real-world outcomes of stage II–III NSCLC patients treated with definitive CRT at a single tertiary care center, with a particular focus on the prognostic role of durvalumab consolidation therapy. By analyzing OS and PFS in relation to established clinical factors such as performance status, comorbidity burden, and PD-L1 expression, this study contributes additional real-world data to the growing evidence on the effectiveness of the PACIFIC regimen in routine clinical practice [13,20,21,22,23].

An important finding is the discrepancy between univariable and multivariable models regarding the effect of durvalumab. While univariable analysis did not demonstrate a significant survival benefit, multivariable modeling—adjusting for confounding variables—revealed durvalumab to be an independent predictor of both prolonged OS (HR 3.99, *p* = 0.008) and PFS (HR 2.81, *p* = 0.023). This underscores the relevance of adjusting for factors such as performance status and comorbidity burden. Indeed, Karnofsky performance status > 80% and CCI ≤ 2 emerged as the strongest independent prognostic markers for OS and PFS. The stepwise AIC-based model selection adds robustness by identifying the most informative predictors with optimal model fit.

Our findings align with and extend the results from the PACIFIC trial, which established the clinical benefit of durvalumab consolidation in unresectable stage III NSCLC. While our observed 3-year OS rate of 38.6% is lower than the 57.0% reported in PACIFIC [12], this is consistent with the heterogeneity and increased vulnerability of real-world populations. Importantly, we observed a median PFS of 20.5 months in durvalumab-treated patients versus 12.0 months in those who did not receive durvalumab, with a 5-year PFS rate of 26.3% across the total cohort.

Similarly, Saad et al. reported significantly prolonged progression-free survival (PFS) and OS with durvalumab compared to historical controls (median OS not reached vs. 24 months, *p* < 0.0001; median PFS 27 vs. 10 months), reinforcing the benefit of immunotherapy beyond clinical trial settings. Their multivariable analysis also confirmed durvalumab as an independent predictor for improved outcomes, consistent with our findings [24].

This is further supported by recent real-world evidence from the Australian PACIFIC-R cohort, reported by Markman, Kao et al., demonstrating a median PFS of 22.4 months and a 3-year OS rate of 59.1% despite heterogeneity in durvalumab initiation timing [25]. Likewise, the Canadian RELEVANCE study by Wheatley-Price, Navani et al. found a median OS of 44.6 months in patients treated with CRT and durvalumab, and observed a clear survival gradient by PD-L1 expression level [26]. These findings affirm that the PACIFIC regimen remains effective in broader clinical contexts.

Real-world data from Mooradian et al. (2024) [27] using a large US database showed even stronger associations: patients treated with durvalumab had significantly longer OS (not reached vs. 19.4 months) and PFS (17.5 vs. 7.6 months), with adjusted hazard ratios of 0.27 and 0.36 for OS and PFS, respectively. These findings underscore the consistent clinical value of durvalumab in diverse populations and healthcare systems [27].

The relationship between PD-L1 expression and durvalumab benefit remains complex. In our cohort, most patients with PD-L1 ≥ 1% received durvalumab, while those with PD-L1 < 1% predominantly did not. In multivariable analysis, PD-L1 ≥ 1% showed a trend toward improved OS (HR 3.72, *p* = 0.063) and was retained in the final PFS model, although it was not a statistically significant predictor. This suggests that PD-L1 status may guide treatment decisions but is not the sole determinant of outcome in this real-world setting.

Importantly, our subgroup analysis combining durvalumab and performance status revealed particularly favorable outcomes in patients with both high KPS and durvalumab treatment. Kaplan–Meier analysis showed markedly improved OS and PFS in this group, reinforcing the importance of performance status as a treatment selection criterion.

This study offers several notable strengths that enhance the validity and clinical relevance of its findings. First, it provides real-world evidence on the effectiveness of durvalumab consolidation therapy in a European cohort, which remains underrepresented in the existing literature. The inclusion of patients treated in routine clinical practice ensures a higher degree of external validity and captures the diversity and complexity of real-life oncologic care, including patients who may not meet the strict eligibility criteria of randomized controlled trials.

Second, the use of comprehensive multivariable Cox regression models—complemented by stepwise variable selection based on Akaike Information Criterion (AIC)—enabled the identification of the most informative prognostic factors while controlling for confounders. This statistical rigor enhances the robustness of the conclusions drawn regarding the impact of durvalumab, Karnofsky performance status, and comorbidity burden.

Third, this study employed a sensitivity analysis adjusted for treatment period (2017–2018 vs. 2019–2022) to account for the phased introduction of durvalumab in clinical practice. This approach strengthens the internal validity by demonstrating that the observed associations are not merely artifacts of temporal treatment variation but represent consistent effects across evolving clinical contexts.

Additionally, the inclusion of bootstrap validation with 1000 iterations further supports the stability and reproducibility of the multivariable model results. This resampling approach reduces the risk of overfitting and confirms the reliability of the key prognostic variables.

Finally, this study’s pragmatic design, which reflects everyday clinical decision-making processes, enhances its translational value. The findings provide clinicians with actionable insights into which patients are most likely to benefit from durvalumab, especially in resource-constrained or heterogeneous patient populations.

Several methodological limitations must be acknowledged. First, the monocentric and retrospective design limits generalizability and introduces potential selection and information bias. Second, durvalumab implementation between 2017 and 2022 was non-uniform, with clinical access and institutional protocols evolving over time. In line with this, our subgroup analysis showed that the majority of patients receiving durvalumab were treated in the later time period (2020–2022), reflecting its increasing adoption in routine care. Sensitivity analyses adjusted for treatment period (to account for durvalumab availability) confirmed the robustness of our findings. Bootstrap-based Cox models revealed consistent survival benefits for CCI ≤ 2, Karnofsky > 80%, and durvalumab use, with the results aligning closely with the main multivariable model. The treatment period itself was not a significant independent predictor of outcome, reinforcing the observed associations despite changing durvalumab accessibility. Treatment allocation was non-randomized, increasing the likelihood of confounding. Additionally, adverse events were not recorded using standardized tools such as CTCAE; therefore, no detailed grading of toxicity severity was possible.

In the present study, a small number of patients with stage II NSCLC and mediastinal lymph node involvement (N2) were included. The decision to incorporate these patients was guided by clinical reasoning, as their disease burden and treatment recommendations aligned more closely with stage III protocols. This reflects real-world practice, where the intensity of therapy is often individualized based on multidisciplinary tumor board recommendations and patient characteristics. Including such cases enhances the external validity of our findings.

The observed discrepancy between univariable and multivariable analyses regarding the effect of durvalumab can be attributed to confounding baseline characteristics. In our cohort, patients with better performance status (KPS > 80%) and lower comorbidity burden (CCI ≤ 2) were more likely to receive durvalumab, which likely biased the unadjusted comparisons. Multivariable modeling allowed us to account for these confounders, revealing durvalumab as an independent predictor of both OS and PFS. This adjustment underscores the importance of controlling for performance-related selection bias in observational studies.

While KPS and CCI consistently emerged as strong prognostic factors in both univariable and multivariable models, we do not interpret them as overshadowing the impact of durvalumab. Rather, these variables help identify patient subgroups who are most likely to benefit from immunotherapy. Our subgroup analyses support this, demonstrating the most favorable outcomes in patients with both good performance status and durvalumab treatment. These findings emphasize the need for thoughtful patient selection and support the integration of functional and comorbidity assessment into clinical decision-making.

Prior real-world analyses suggest that consolidation durvalumab can be effective in diverse patient populations. In this context, the study by Park, Hong et al. (2024) [28] offers important insights into the role of durvalumab in elderly patients—a population often underrepresented in clinical trials. In their cohort of 286 patients, 42% were aged ≥70 years. Notably, survival outcomes in elderly patients were comparable to those in younger individuals, with no significant differences in median PFS (17.7 vs. 19.4 months; *p* = 0.43) or median OS (35.7 months vs. not reached; *p* = 0.13). These findings suggest that advanced age alone should not preclude patients from receiving durvalumab after CRT [28].

In our cohort, the median overall survival was 25.5 months in patients aged < 70 years compared to 15.0 months in those aged ≥ 70 years. Median progression-free survival was 14.0 months in both age groups. The estimated 3-year and 5-year PFS rates were 30.4% and 22.5% for patients under 70 years, and 31.1% at both timepoints for those aged 70 or older. These findings suggest that advanced age alone should not preclude patients from receiving durvalumab after CRT, especially when carefully selected and monitored.

The findings of Lau, Ryan et al. (2021) further support the safety and efficacy of durvalumab consolidation in elderly patients with stage III NSCLC [29]. In their retrospective analysis of 115 patients—44 of whom were aged ≥ 70 years—the completion rates for CRT and chemotherapy dose intensity were similarly high in both age groups, indicating that older patients can tolerate definitive CRT comparably to younger cohorts. Importantly, the majority of elderly patients without disease progression after CRT received durvalumab (78%), with a low incidence of grade ≥ 3 immune-related adverse events (9%), which was not significantly different from younger patients (6%).

Our data support this perspective, highlighting that with appropriate selection, elderly patients may experience comparable progression-free outcomes and derive meaningful clinical benefit from durvalumab consolidation therapy. However, age-related differences in overall survival underscore the importance of individualized risk–benefit assessment.

However, the increased frequency of treatment-related adverse events (AEs) in elderly patients observed in that study underscores the importance of careful patient selection and AE monitoring. Specifically, grade 3/4 AEs, treatment-related deaths, and durvalumab discontinuation due to pulmonary toxicity were significantly more common in the elderly subgroup. This aligns with our own findings, where adverse events were prevalent in both groups, but a standardized grading of severity was not available, limiting detailed comparison.

## 5. Conclusions

This retrospective real-world study of stage III NSCLC patients treated with chemoradiotherapy confirms the prognostic importance of performance status, comorbidity burden, and durvalumab treatment. Higher Karnofsky scores (>80%), lower CCI (≤2), and receipt of durvalumab since its availability in 2019 were independently associated with improved overall survival, consistent with findings from clinical trials like PACIFIC.

Although durvalumab did not significantly impact PFS in univariable analysis, multivariable modeling revealed favorable trends. PD-L1 ≥ 1% was associated with improved survival, though without statistical significance.

These findings support the integration of durvalumab into routine care for fit patients and highlight the need for further prospective research to refine patient selection and optimize real-world outcomes in stage III NSCLC. Sensitivity analyses adjusting for treatment period confirmed the robustness of the multivariable results, indicating that the observed benefits of durvalumab persist even when accounting for the fact that durvalumab was not available at all prior to 2019.

## Figures and Tables

**Figure 1 cancers-17-02498-f001:**
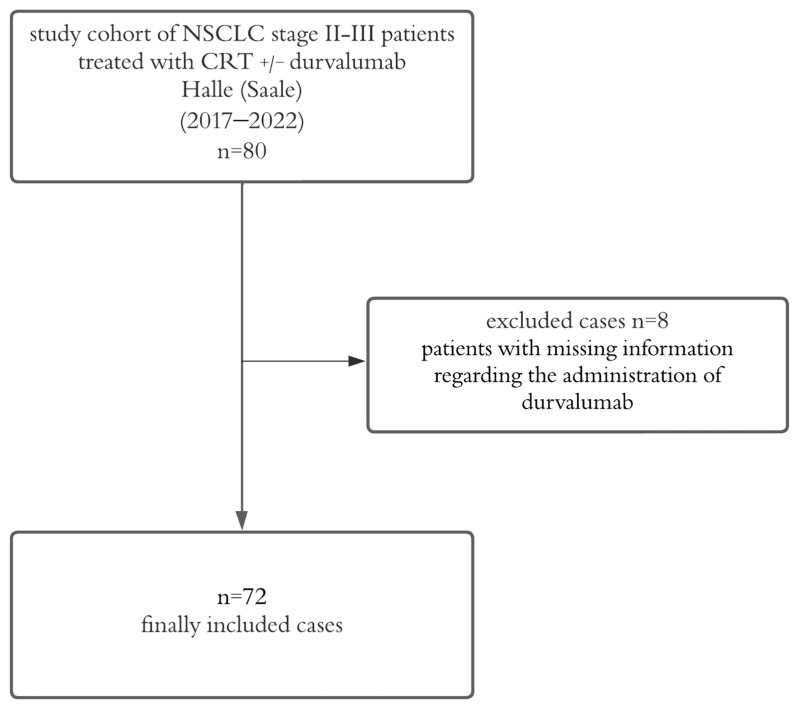
Patient selection flowchart. Flow diagram illustrating the selection of the final study cohort.

**Figure 2 cancers-17-02498-f002:**
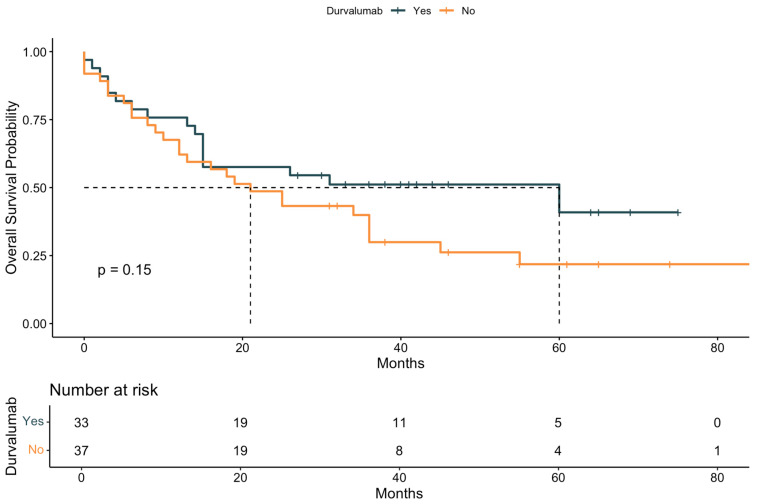
Kaplan–Meier curve for overall survival (OS) stratified by durvalumab treatment. Survival probability is plotted over time (in months) for patients who did or did not receive durvalumab following definitive chemoradiotherapy. Median survival, 95% confidence intervals, and the number at risk are shown for each time point. The *p*-value was calculated using the log-rank test. The dashed line indicates the median survival of both groups.

**Figure 3 cancers-17-02498-f003:**
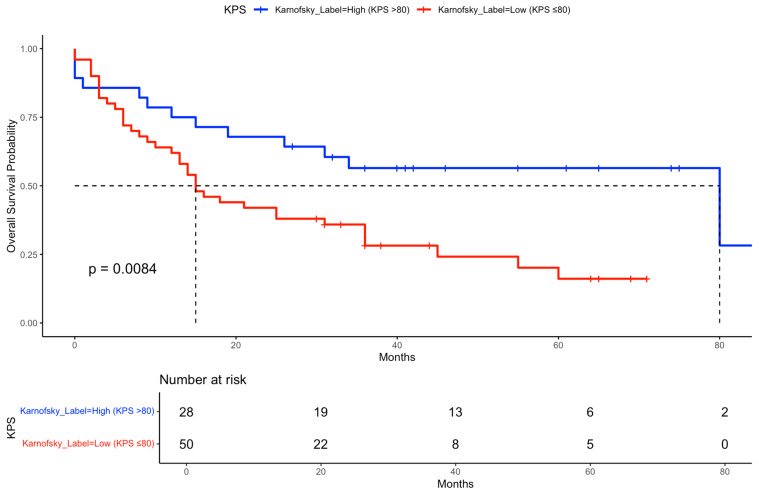
Kaplan–Meier curve for OS stratified by Karnofsky performance status. Overall survival probability is shown for patients grouped by Karnofsky performance status (≤80% vs. >80%). Median survival times and the number of patients at risk over time are indicated for each group. The dashed line indicates the median survival of both groups.

**Figure 4 cancers-17-02498-f004:**
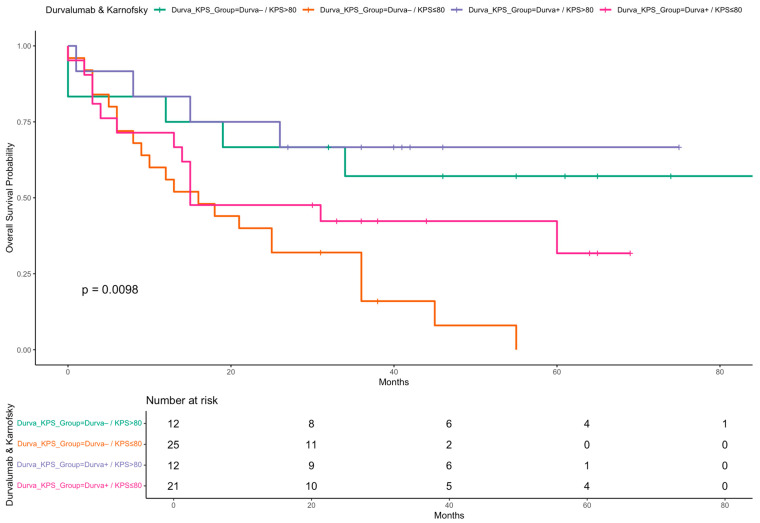
Kaplan–Meier curve for OS stratified by durvalumab treatment and Karnofsky performance status. Overall survival probabilities are shown for four patient subgroups defined by receipt of durvalumab (Yes/No) and Karnofsky performance status (KPS ≤ 80 vs. >80). Survival time is shown in months. A log-rank test was used to compare survival distributions across groups.

**Figure 5 cancers-17-02498-f005:**
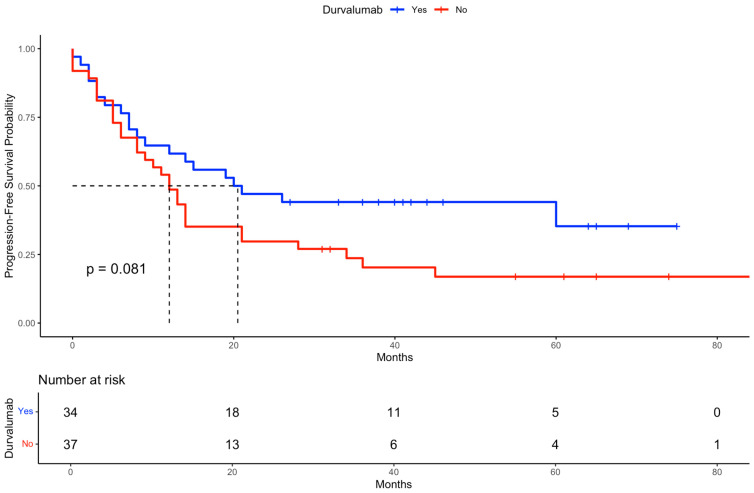
Kaplan–Meier curve for progression-free survival (PFS) stratified by durvalumab treatment. The figure shows the PFS probability over time for patients who received durvalumab (blue curve) compared to those who did not (red curve). The log-rank test was used to evaluate differences between groups. The dashed line indicates the median survival of both groups.

**Figure 6 cancers-17-02498-f006:**
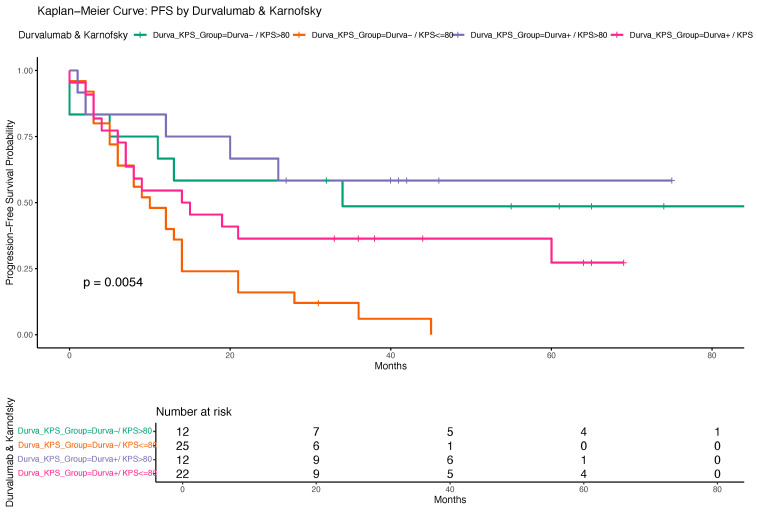
Kaplan–Meier curve for progression-free survival (PFS) stratified by durvalumab treatment and Karnofsky performance status. The figure illustrates PFS probabilities for four patient subgroups defined by the combination of durvalumab treatment (yes/no) and Karnofsky performance status (≤80 vs. >80). Differences in PFS between groups were assessed using the log-rank test. Survival times are displayed in months.

**Table 1 cancers-17-02498-t001:** Charlson Comorbidity Index (CCI) calculation chart adapted after the work of Charlson et al. ^1^ HIV = human immunodeficiency virus; ^2^ AIDS = acquired immunodeficiency syndrome [19].

Comorbidity	Weight	Criteria
Myocardial Infarction	1	History of myocardial infarction or coronary artery disease
Congestive Heart Failure	1	History of heart failure
Peripheral Vascular Disease	1	Claudication, peripheral artery disease, or previous vascular surgery
Cerebrovascular Disease	1	Stroke, transient ischemic attack, or history of cerebral hemorrhage
Dementia	1	Clinical diagnosis of dementia
Chronic Pulmonary Disease	1	Chronic obstructive pulmonary disease or asthma
Connective Tissue Disease	1	Rheumatoid arthritis or systemic lupus erythematosus
Peptic Ulcer Disease	1	History of gastric or duodenal ulcer
Mild Liver Disease	1	Chronic liver disease without liver failure (e.g., cirrhosis without ascites)
Diabetes without Complications	1	Diabetes mellitus without end-organ damage
Diabetes with Complications	2	Diabetes with end-organ damage, such as retinopathy or nephropathy
Hemiplegia or Paraplegia	2	Paralysis due to stroke, spinal cord injury, or other causes
Moderate or Severe Renal Disease	2	Chronic kidney disease with creatinine > 3 mg/dL or dialysis-dependent
Cancer (Non-Metastatic, Active Treatment)	2	Any solid tumor without metastasis, currently treated
Leukemia	2	Chronic or acute leukemia
Lymphoma	2	Non-Hodgkin or Hodgkin lymphoma
Moderate or Severe Liver Disease	3	Cirrhosis with complications (e.g., ascites, encephalopathy)
Metastatic Solid Tumor	6	Any solid tumor with metastasis
AIDS ^2^/HIV ^1^	6	HIV ^1^ infection with AIDS ^2^ or opportunistic infections

**Table 2 cancers-17-02498-t002:** Baseline characteristics of patients receiving chemoradiotherapy with and without durvalumab.

Characteristic	Overall n = 72	CRT with Durvalumab n = 35	CRT Without Durvalumab n = 37
**Age at diagnosis** (years) ^1^	66 (46, 83)	66 (46, 83)	67 (50, 82)
**Sex**			
male	54 (75%)	26 (74%)	28 (76%)
female	18 (25%)	9 (26%)	9 (24%)
**Karnofsky Index** (%) ^2^	80 (50, 100)	80 (70, 100)	80 (50, 100)
**PD-L1 status** (%) ^3^	5 (1, 60)	30 (5, 75)	1 (1, 2)
**T-stage**			
1	2 (2.9%)	2 (5.9%)	0 (0%)
2	8 (11%)	3 (8.8%)	5 (14%)
3	19 (27%)	7 (21%)	12 (33%)
4	41 (59%)	22 (65%)	19 (53%)
**N-stage**			
0	14 (20%)	7 (20%)	7 (19%)
1	12 (17%)	6 (17%)	6 (17%)
2	27 (38%)	13 (37%)	14 (39%)
3	18 (25%)	9 (26%)	9 (25%)
**Histology**			
adenocarcinoma	20 (29%)	8 (23%)	12 (34%)
large-cell carcinoma	1 (1.4%)	1 (2.9%)	0 (0%)
neuroendocrine carcinoma	1 (1.4%)	0 (0%)	1 (2.9%)
squamous cell carcinoma	48 (69%)	26 (74%)	22 (63%)
**Chemotherapy**			
Carboplatin monotherapy	4 (5.6%)	3 (8.6%)	1 (2.7%)
Carboplatin/Paclitaxel	25 (35%)	10 (29%)	15 (41%)
Cisplatin/Vinorelbine	43 (60%)	22 (63%)	21 (57%)
**Total radiation dose** (Gy) ^4^	66.00, 65.17 (60.00, 66.00)	66.00, 65.00 (60.00, 66.00)	66.00, 65.33 (60.00, 66.00)

^1^ Mean (Min, Max); n (%); ^2^ Median (Min, Max); ^3^ Median (Q1, Q3); ^4^ Median, Mean (Min, Max). Continuous variables are presented as median (minimum, maximum) or mean (minimum, maximum), as appropriate. Categorical variables are presented as counts (percentages). The “Overall” column includes all patients regardless of treatment group. CCI = Charlson Comorbidity Index; CRT = chemoradiotherapy; PD-L1 = programmed death-ligand 1; UICC = Union for International Cancer Control.

**Table 3 cancers-17-02498-t003:** Univariable and multivariable Cox regression analyses for overall survival (OS). Hazard ratios (HR) with 95% confidence intervals (CI) and *p*-values are reported for each variable. Univariable models were calculated separately for each covariate using the entire available cohort. The multivariable model was derived using stepwise selection based on the Akaike Information Criterion (AIC) and includes covariates with the strongest independent associations with OS.

Characteristic	Univariate			Multivariate (Stepwise)		
	HR	95% CI	*p*-Value	HR	95% CI	*p*-Value
**Age at diagnosis**	1.01	0.98, 1.05	0.4			
**Sex**			0.049			
male	—	—				
female	0.48	0.21, 1.06				
**CCI-group**			0.2			0.009
>2	—	—		—	—	
≤2	0.65	0.35, 1.21		0.39	0.18, 0.82	
**Karnofsky Index**			0.007			0.003
≤80%	—	—		—	—	
>80%	0.42	0.22, 0.82		0.29	0.12, 0.69	
**T-status**			>0.9			
T1–2	—	—				
T3–4	1.04	0.50, 2.17				
**N-status**			0.4			0.093
0	—	—		—	—	
1	1.64	0.59, 4.59		0.55	0.13, 2.29	
2	1.94	0.82, 4.60		2.12	0.73, 6.16	
3	1.17	0.45, 3.09		0.97	0.29, 3.27	
**durvalumab**			0.12			0.008
yes	—	—		—	—	
no	1.65	0.88, 3.09		3.99	1.48, 10.8	
**PD-L1 status**			0.7			0.063
<1%	—	—		—	—	
≥1%	1.15	0.58, 2.25		3.72	1.27, 10.9	
Unknown	1.45	0.56, 3.75		1.98	0.49, 8.02	

Abbreviations: CI = confidence interval, HR = hazard ratio.

**Table 4 cancers-17-02498-t004:** Univariable and multivariable Cox regression analyses for progression-free survival (PFS). Hazard ratios (HRs), 95% confidence intervals (CIs), and *p*-values are presented for each variable. The univariable model includes all relevant clinical and biological parameters. The multivariable model was selected using stepwise variable selection based on Akaike’s Information Criterion (AIC).

Characteristic	Univariable			Multivariable (Stepwise)		
	HR	95% CI	*p*-Value	HR	95% CI	*p*-Value
**Age at diagnosis**	1.00	0.96, 1.03	0.8			
**Sex**			0.2			
male	—	—				
female	0.63	0.30, 1.29				
**CCI-group**			0.3			0.048
>2	—	—		—	—	
≤2	0.73	0.40, 1.34		0.53	0.28, 1.01	
**Karnofsky Index**			0.003			<0.001
≤80	—	—		—	—	
>80	0.37	0.19, 0.76		0.29	0.14, 0.60	
**T-status**			0.5			
T1–2	—	—				
T3–4	1.36	0.58, 3.21				
**N-status**			0.4			
0	—	—				
1	1.45	0.54, 3.87				
2	1.93	0.85, 4.37				
3	1.19	0.48, 2.97				
**durvalumab**			0.2			0.023
yes	—	—		—	—	
no	1.52	0.86, 2.71		2.81	1.22, 6.47	
**PD-L1 status**			0.8			0.039
<1	—	—		—	—	
≥1	0.88	0.47, 1.65		1.70	0.72, 4.04	
unknown	0.73	0.27, 2.02		0.40	0.14, 1.14	

Abbreviations: CI = confidence interval, HR = hazard ratio.

## Data Availability

The datasets and materials used and/or analyzed during the current study are available from the corresponding author on reasonable request.

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
