# Peer review of "Real-World Outcomes of Chemoradiotherapy in Patients with Stage II/III Non-Small-Cell Lung Cancer in the Durvalumab Era: An Observational Study"

_cancers, 2025, doi:10.3390/cancers17152498_

Round 1
Reviewer 1 Report
Comments and Suggestions for Authors
This is a good real-life study on lung cancer patients.
Here are my comments:
- The abstract needs to be summarized. Look at what the journal says.
- The introduction could explain a little about what Durvalumab is.
- Line 113: explain what the abbreviation means.
- Lines 162 to 165 have different fonts. Check this because it happens again in the manuscript.
- Figure 1: better quality.
- The tables are too large. Include the most relevant information and put the rest in the supplement.
- I would put the limitations of the study at the end of the discussion and include the strengths.
Author Response
|
1. Summary |
|
|
|
Thank you very much for taking the time to review this manuscript. Please find the detailed responses below and the corresponding revisions/corrections highlighted/in track changes in the re-submitted files.
|
||
|
2. Questions for General Evaluation |
Reviewer’s Evaluation |
Response and Revisions |
|
Does the introduction provide sufficient background and include all relevant references? |
Can be improved |
|
|
Are all the cited references relevant to the research? |
Can be improved |
|
|
Is the research design appropriate? |
Yes |
|
|
Are the methods adequately described? |
Yes |
|
|
Are the results clearly presented? |
Can be improved |
|
|
Are the conclusions supported by the results? |
Yes |
|
|
3. Point-by-point response to Comments and Suggestions for Authors |
||
|
Comments 1:
|
||
|
Comment: This is a good real-life study on lung cancer patients. Thank you for your positive assessment and for taking the time to review our manuscript.
Comment: The abstract needs to be summarized. Look at what the journal says. Response 1: Thank you for pointing this out. We agree with this comment. Therefore, we have added a 'Simple Summary' in accordance with the journal’s formatting requirements and streamlined the abstract to improve clarity.
Comment: The introduction could explain a little about what Durvalumab is. Response 2: Agree. We have added a brief explanation of what Durvalumab is in the Introduction to enhance understanding.
Comment: Line 113: explain what the abbreviation means. Response 3: Thank you for this helpful comment. We have reviewed and revised all abbreviations throughout the manuscript to ensure that they are clearly defined at their first mention.
Comment: Lines 162 to 165 have different fonts. Check this because it happens again in the manuscript. Response 4: Thank you. We have carefully checked the formatting and have corrected all inconsistencies in font throughout the manuscript.
Comment: Figure 1: better quality. Response 5: Thank you for this suggestion. Figure 1 has been replaced with a high-resolution version (300 dpi) to ensure better quality.
Comment: The tables are too large. Include the most relevant information and put the rest in the supplement. Response 6: Thank you for the recommendation. We have shortened Table 2 (Patient Characteristics) and moved the full version to the Supplement. In addition, Table 5 was also moved to the Supplement. Regression tables were retained in full to transparently present the statistical models.
Comment: I would put the limitations of the study at the end of the discussion and include the strengths. Response 7: Thank you. We have moved the limitations section to the end of the Discussion and added a new section explicitly discussing the strengths of the analysis.
|
||
|
4. Response to Comments on the Quality of English Language |
||
|
Point 1: |
||
|
The English is fine and does not require any improvement. |
||

Reviewer 2 Report
Comments and Suggestions for Authors
I would like to commend Drs Andreas, Jonas, et al., on their well designed and completed study. The main objective of this study was to compare the real world impact of adjuvant durvalumab in patients with stage II and III non small cell lung cancer definitively managed with chemoradiation therapy. It is well established that patients benefit from adjuvant durvalumab treatment following definitive chemoradiation therapy for unresectable stage III non small cell lung cancer following the PACIFIC trial, and this study looked to further investigate that by evaluating the effects in a real world cohort of patients. The paper evaluated 72 patients, at a single center, between 2017 and 2022, who underwent definitive chemoradiation therapy for stage III non small cell lung cancer and were then treated with or without adjuvant durvalumab. They concluded that there was an improvement in overall survival and progression free survival in those patients with better Karnofsky performance status (KPS), Charlson comorbidity index (CCI) and those who were treated with adjuvant durvalumab.
This study represents a retrospective review of the treatment of stage III lung cancer with definitive chemoradiation therapy followed by adjuvant durvalumab versus not in a single center. The author’s findings of differences in survival, and disease free survival based on preoperative factors such as KPS and CCI is well supported by prior literature, additionally the finding of improved survival with adjuvant durvalumab is also well supported. The conclusions of this paper, mirror those in other adjuvant immunotherapy studies following definitive chemoradiation therapy. The data presented in this study support the overall conclusions, and I surmise that as more immunotherapy options come on the market even more questions will arise as to the optimal treatment of patients following definitive chemoradiation therapy. This study helps to lay some of the framework for that discussion.
There are several potentially significant biases to a retrospective study, especially one conducted at a single institution, and the authors comment on this fact and that the number of patients is fairly small, and therefore the study has limited power, which may obscure the overall impact of the findings.
I wonder if the authors could comment more on the choice to include stage II patients in this evaluation even though they have significantly improved survival compared to stage III and the title of the article implies that only patients with stage III disease are included.
I found it interesting that the median OS was worse with the durvalumab group, however the progression free survival almost double that of those that did not receive the immunotherapy. I wonder if the authors could surmise how the progression and overall survival are so different. The lack of univariate difference between receiving durvalumab and not is interesting, and I wonder if the authors could discuss this in more detail, and make some statements as to why they would think that there is an uncovering of a significant effect when it is included in the multivariate analysis.
It seems that the primary impactors on survival in the univariate and multivariate assessments are the KPS and CCI, and I wonder if this is the primary drivers of survival in the study rather than the impact of durvalumab.
Overall, I enjoyed reading this well written article, and I would like to thank the authors for their manuscript and contribution to the treatment of this challenging disease process.
Author Response
|
Response to Reviewer 2 Comments
|
||
|
1. Summary |
|
|
|
Thank you very much for taking the time to review this manuscript. Please find the detailed responses below and the corresponding revisions/corrections highlighted/in track changes in the re-submitted files.
|
||
|
2. Questions for General Evaluation |
Reviewer’s Evaluation |
Response and Revisions |
|
Does the introduction provide sufficient background and include all relevant references? |
Yes |
|
|
Are all the cited references relevant to the research? |
Yes |
|
|
Is the research design appropriate? |
Yes |
|
|
Are the methods adequately described? |
Yes |
|
|
Are the results clearly presented? |
Yes |
|
|
Are the conclusions supported by the results? |
Yes |
|
|
3. Point-by-point response to Comments and Suggestions for Authors |
||
|
Comments:
|
||
|
I wonder if the authors could comment more on the choice to include stage II patients in this evaluation even though they have significantly improved survival compared to stage III and the title of the article implies that only patients with stage III disease are included. |
||
|
Response: |
||
|
Response: Thank you very much for your thoughtful and constructive comments. We appreciate the opportunity to clarify the points raised. |
||
|
I wonder if the authors could comment more on the choice to include stage II patients in this evaluation even though they have significantly improved survival compared to stage III and the title of the article implies that only patients with stage III disease are included. |
||
|
Response: |
||
|
Response: Thank you very much for your thoughtful and constructive comments. We appreciate the opportunity to clarify the points raised. |
||

Reviewer 3 Report
Comments and Suggestions for Authors
Thank you very much for allowing me to review this article.
I think this review is an article that is redundant with other phase IV facts or real world data already previously reported by the laboratory itself or by other scientific communities or hospitals:
*An Open-Label, Multi-Center, Global Study to Evaluate Long Term Safety and Efficacy in Patients Who are Receiving or Who Previously Received Durvalumab in Other Protocols (WAVE). A phase 4 owith 163 patients enrolled.
Study Start Date: 05 Sept 2019
Primary Completion Date: 31 Oct 2022
Study Completion Date: 31 Oct 2024
*Real-world survival outcomes, treatment patterns, and impact of PD-L1 expression among patients with unresectable, stage III NSCLC treated with CRT → durvalumab in Canada: The RELEVANCE study
https://doi.org/10.1016/j.lungcan.2025.108583
Durvalumab is approved in some countries for patients with positive PDL1. I think this factor should have been sub-stratified from the outset (I don't see it as the case), and, furthermore, it is described in the results, but not really clear, direct, or priority.
Besides, there are some ideas that perhaps need some modifications:
*Introduction, lines 48-.....-93:. There are many more later studies based on real-time data or phase 4 that I think should be named apart from the Pacific, please reconsider this point.
*2. Methods 94 2.1. Data and Material: Line 100: All patients with stage II–III non-small cell and Lines 116-119: They are talking about the same topic (stage) but it is separate, please consider reorganizing the introduction so that the same topics are together.
*3. Results. 3.2 Patient characteristics, lines 198-200: “These findings suggest a gradual implementation of durval-198 umab following its approval, with increasing alignment between PD-L1 expression and 199 treatment choice in later years”. I think this comment is more suitable as a discussion, given that a suggestive interpretation of the findings is being made.
*4. Discussion: lines 366-367 “PD-L1 expression, this study contributes additional real-world data to the growing evidence on the effectiveness of the PACIFIC región. I think it would be appropriate to make a more critical comment given that the statistical study does not have sufficient power per sample size to assess overall survival results according to PDL1 as you said in the article.
Author Response
|
1. Summary |
|
|
|
Thank you very much for taking the time to review this manuscript. Please find the detailed responses below and the corresponding revisions/corrections highlighted/in track changes in the re-submitted files.
|
||
|
2. Questions for General Evaluation |
Reviewer’s Evaluation |
Response and Revisions |
|
Does the introduction provide sufficient background and include all relevant references? |
Can be improved |
|
|
Are all the cited references relevant to the research? |
Can be improved |
|
|
Is the research design appropriate? |
Yes |
|
|
Are the methods adequately described? |
Yes |
|
|
Are the results clearly presented? |
Yes |
|
|
Are the conclusions supported by the results? |
Yes |
|
|
3. Point-by-point response to Comments and Suggestions for Authors |
||
|
|
||
|
Comment 1: There are many more later studies based on real-time data or phase 4 that I think should be named apart from the PACIFIC, please reconsider this point. |
||
|
|
||
|
|
||
|
|
||
|
|
||
|
Comment 1: There are many more later studies based on real-time data or phase 4 that I think should be named apart from the PACIFIC, please reconsider this point. |
||
|
|
||
|
|
||

Round 2
Reviewer 2 Report
Comments and Suggestions for Authors
I would like to commend the authors for their revised manuscript, which addresses the primary concerns that I had with the first iteration. Thank you